# Developments in Extracellular Matrix-Based Angiogenesis Therapy for Ischemic Heart Disease: A Review of Current Strategies, Methodologies and Future Directions

**DOI:** 10.3390/biotech14010023

**Published:** 2025-03-19

**Authors:** Jad Hamze, Mark Broadwin, Christopher Stone, Kelsey C. Muir, Frank W. Sellke, M. Ruhul Abid

**Affiliations:** Division of Cardiothoracic Surgery, Cardiovascular Research Center, Rhode Island Hospital, Warren Alpert Medical School of Brown University, Providence, RI 02903, USA

**Keywords:** Extracellular Matrix (ECM), myocardial regeneration, bioengineered biomaterials, ischemic heart disease, coronary angiogenesis

## Abstract

Ischemic heart disease (IHD) is the leading cause of mortality worldwide, underscoring the urgent need for innovative therapeutic strategies. The cardiac extracellular matrix (ECM) undergoes extreme transformations during IHD, adversely influencing the heart’s structure, mechanics, and cellular signaling. Researchers investigating the regenerative capacity of the diseased heart have turned their attention to exploring the modulation of ECM to improve therapeutic outcomes. In this review, we thoroughly examine the current state of knowledge regarding the cardiac ECM and its therapeutic potential in the ischemic myocardium. We begin by providing an overview of the fundamentals of cardiac ECM, focusing on the structural, functional, and regulatory mechanisms that drive its modulation. Subsequently, we examine the ECM’s interactions within both chronically ischemic and acutely infarcted myocardium, emphasizing key ECM components and their roles in modulating angiogenesis. Finally, we discuss recent ECM-based approaches in biomedical engineering, focusing on different types of scaffolds as delivery tools and their compositions, and conclude with future directions for therapeutic research. By harnessing the potential of these emerging ECM-based therapies, we aim to contribute to the development of novel therapeutic modalities for IHD.

## 1. Introduction

Ischemic heart disease (IHD) is the leading cause of death worldwide, accounting for over 16% of global mortality [1]. Although surgical and percutaneous revascularization are considered standard treatments for severely diseased myocardium, a considerable number of patients are not suitable for such procedures [2]. Moreover, even patients eligible for such interventions may encounter secondary complications, such as ischemia-reperfusion injury, which prevents the full restoration of the structure and function within the necrotic myocardium [3,4]. Due to the paucity of effective treatment options for IHD, there is a significant demand for alternative non-surgical and more effective therapies.

Acute myocardial infarction (AMI) results in a substantial loss of cardiomyocytes, resulting in remodeling of the myocardium and posing significant challenges in the recovery of cardiac function [5,6]. Due to the limited regenerative capacity of adult cardiomyocytes, the infarcted myocardium is replaced with scar tissue composed primarily of extracellular matrix (ECM), which provides structural support to a certain degree but fails to restore the heart’s functional capacity [7]. The ECM is mostly composed of collagen and elastic fibers dispersed in proteoglycans, glycoproteins, and glycosaminoglycans. The cumulative effect of these alterations in cardiac composition precipitates changes in the structure, mechanics, and regulation of cellular responses in the heart. The process of infarct healing can be divided into three main phases—inflammatory, proliferative, and maturation—each defined by changes within the ECM [8]. The inflammatory phase involves the activation of immune cells by disrupted ECM fragments. This triggers the proliferative phase, during which fibroblasts differentiate into myofibroblasts and neovascularization begins [9]. During these phases, fibroblasts degrade the matrix and replace it with structural ECM proteins, such as collagen, while also differentiating into myofibroblasts that secrete contractile proteins [10]. ECM crosslinking by lysyl oxidase and hydroxylase that occurs during maturation provides essential structural stability after infarction; however, such stability comes at the expense of a marked depreciation in multiple parameters of cardiac function and may result in heart failure [11,12].

The appreciation of the critical role of the ECM in cardiac healing has positioned it as a focal point in the design of regenerative therapies aimed at ameliorating cardiac dysfunction following injury. In this review, we describe the current state of knowledge regarding the therapeutic potential of ECM in the context of multiple ischemic myocardial disease states. We begin by providing an overview of the fundamental role of the ECM, including its functional, structural, and regulatory mechanisms, in healthy and diseased hearts. We thoroughly examine the alterations in the ECM imposed by both acute and chronic myocardial ischemia, emphasizing specific ECM components and their roles in tissue regeneration. Finally, we discuss ECM in the realm of biomedical engineering, focusing on different types of scaffolds as ECM delivery tools and the biomaterial bases comprising them, and conclude with future directions for ECM in therapeutic research.

## 2. The ECM in Normal Cardiac Physiology

The ECM plays an important role in the heart by influencing essential cellular processes during organ development and regulating a viable microenvironment for cardiomyocytes [13,14]. The structural composition of the ECM can be divided into three main categories. The epimysium, located on the endocardial and epicardial surfaces, provides support for the endothelial and mesothelial cells that comprise these structures. The perimysium, by virtue of its myocardial location, surrounds and connects muscle fibers. Finally, the endomysium arises from the perimysium and encapsulates muscle fibers to provide a linkage to cardiomyocyte cytoskeletal proteins across plasma membranes [15,16]. Morphologically, the ECM is comprised of a complex laminar network of fibrillary collagens and non-fibrillary structural components that exert both signaling and architectural functions. The fibrillar collagenous matrix is comprised mostly of type I and type III collagens anchored to non-fibrillary components, such as basement membranes, proteoglycans, and glycoproteins, through collagen type IV and fibronectin. In addition, the ECM contains a large reservoir of anchored growth factors, cytokines, chemokines, proteases, protease inhibitors, and noncoding RNAs [17,18].

The elements of the cardiac ECM are dynamic and have the capacity to continuously remodel, by which they can respond to physiological changes. This process is facilitated by the generation of bioactive peptides, known as matrikines, through the enzymatic degradation of ECM macromolecules [13,19]. Matrikine-specific receptors are involved in a diverse range of pathways essential for maintaining a viable microenvironment, including ECM renewal, cellular proliferation, immune responses, organ development, wound repair, and angiogenesis [20].

## 3. The ECM in the Chronically Ischemic Heart

Chronic ischemic cardiomyopathy is a multifarious detriment to effective ECM function. One mechanism by which this occurs is the development of myocardial interstitial fibrosis (MIF), a state characterized by excessive deposition of collagen within the myocardium and concomitant activation of a matrix metalloproteinase (MMP) expression profile that reinforces this state [21,22,23,24,25]. MMPs serve as essential regulators of both inflammatory and reparative cascades by facilitating the proteolytic processing of cytokines, chemokines and growth factors [7,8]. When activated in the inflammatory environment of chronic myocardial ischemia, MMPs serve as degradative enzymes [26]. For instance, in a porcine model of chronic ischemia, in which the left anterior descending coronary artery was occluded for 6 h followed by 3 h of reperfusion, immunoblotting demonstrated the upregulation of collagenase and gelatinase family members—MMP-1, MMP-2, and MMP-9, all significant contributors to ECM fragmentation—within the ischemic myocardium [27]. Furthermore, metabolic syndrome (MS), which is often comorbid with chronic myocardial ischemia, appears to act through alterations in enzymes such as MMP-9 to exacerbate these changes. A study in patients with MS demonstrated that elevated plasma MMP-9 levels were independently associated with MS and correlated with increased cardiovascular risk, suggesting a link between high MMP-9 levels and poor CV outcomes [28]. An additional domain of ischemia-induced ECM alterations arises from the proliferation of the matricellular proteins responsible for the regulation of diverse cellular functions [29]. These proteins include cysteine (SPARC), thrombospondins (TSP)-1, -2, and -4, tenascin-C and -X, periostin, osteopontin, and CCN (cystein-rich protein 61/ connective tissue growth factor) [29]. Matricellular proteins interact with both structural matrix proteins and cell surface receptors, facilitating the transduction of cytokine and growth factor signals and the modulation of proteases and other bioactive mediators [30].

## 4. The ECM in the Acutely Infarcted Heart

Due to the heart’s limited natural regenerative ability, the acute loss of cardiomyocytes following non-revascularized infarction leads to a substantial decline in cardiac function [6]. The natural history of the infarcted myocardium thereafter entails the replacement of necrotic tissue with a matrix-based scar involving, as mentioned above, the three overlapping phases of inflammation, proliferation, and maturation [31]. Synchronously, non-infarcted areas undergo additional remodeling in response to hemodynamic loading and the accumulation of immune cells and fibroblasts imposed by the injured region [32]. During the inflammatory phase, ECM fragments, following activation by proteolytic cleavage, are transduced by immune cells to trigger an inflammatory response. This phase also involves the production of a provisional matrix comprised of plasma proteins, which promote the influx of reparative and immune cells. The proliferative phase is characterized by the phagocytosis of dead cells and the extracellular matrix, ultimately suppressing inflammation. Myofibroblasts are activated by signaling cascades initiated during this process to initiate the limited formation of immature neovessels. Matricellular proteins are the non-structural proteins found in cardiac ECM which facilitate the spatial regulation of growth factor signaling and often impose a check on the fibrotic response comprised of the secretion of structural collagens by activated fibroblasts. As the infarct matures, crosslinking of these collagens yields a detrimental increase in ventricular stiffness, with ongoing deposition culminating in the progression to heart failure. Throughout this process, cardiomyocytes, fibroblasts, and immune cells in the ischemic myocardium synthesize and release large quantities of collagenases, gelatinases, and cathepsins, all of which contribute to ECM fragmentation [27,33,34]. This proteolytic activity is regulated by tissue inhibitors of metalloproteinases (TIMPs), which control the activity of MMPs and other ECM-degrading enzymes to prevent excessive matrix degradation or fibrosis [35].

## 5. ECM Components as Regulators of Cardiac Angiogenesis

The myocardial ECM provides support to the angiogenic process by providing anchorage and structural stabilization for sprouting vessels, and serving as a reservoir for associated cytokines and growth factors [36,37]. In this section, we enumerate the factors that comprise and foster the cardiac ECM and describe their roles in cardiac angiogenesis, with Table 1 providing a comprehensive overview of each regulator in cardiac angiogenesis.

***Endostatin*:** Derived from the precursor human collagen molecule type XVIII, endostatin is an inhibitor of angiogenesis that exerts this function by antagonizing the migration and proliferation of endothelial cells [38]. The effect of endostatin in patients with coronary artery disease (CAD) has been characterized by research that revealed a significant upregulation of endostatin protein levels in the ischemic heart, which was correlated with diminished angiogenesis and the formation of poorly developed collaterals [39,40] (Figure 1).

***Vascular Endothelial Growth Factor (VEGF):*** Among the various cardiac angiogenic modulators in the ECM, VEGF stands out as one of the most important in the development and differentiation of the vascular network, as there is a wealth of evidence demonstrating that VEGF significantly increases perfusion and improves tissue metabolism, cardiac function, and cardiac protection in the ischemic myocardium [13,41]. A study utilizing a left anterior descending coronary artery ligation model in rats showed significant angiogenic effects of VEGF165, the most abundant isoform of VEGF, via an intramyocardial injection within a hydrogel model. In that study, both collagen content and cell apoptosis were decreased, whereas VEGF165 expression and arterial and capillary densities within the ischemic myocardium were increased [42]. The mechanism of VEGF in angiogenesis is wide-ranging. During inflammation, VEGF is released by matrix metalloproteinases, plasmin, urokinase-type plasminogen activator, elastase, and tissue kallikrein. *These proteases cleave VEGF, which can have dual effects depending on the context. In carcinogenesis, cleavage may release bioactive VEGF fragments that enhance angiogenesis, while in other conditions, degradation of VEGF can reduce its availability and suppress angiogenic signaling.* Broadly, these mechanisms establish the ability of VEGF to promote blood vessel growth and remodeling while also providing survival and mitogenic stimuli for ECs [43] (Figure 1).

***Placental Growth Factor (PlGF):*** PlGF exists within the VEGF family of cytokines, and its principal function is to control vascular and lymphatic endothelial development. Its angiogenic capabilities are demonstrated by PlGF’s ability to increase the tyrosine kinase activity of VEGF, which activates a signaling pathway leading to the formation of new blood vessels [44]. PlGF also plays a significant role in the production of nitric oxide via its regulation of the eNOS enzyme. Additionally, inflammation and oxidative stress contribute to PlGF expression [45]. In a recent study, which investigated PlGF’s role in cardiomyogenesis and vasculogenesis, single-cell RNA sequencing of human and primate hearts was employed and found that PlGF is expressed in second heart field progenitors at early stages of development, and later in smooth muscle cells (SMCs) and ECs. To study the effects of PlGF on cardiomyocytes (CM), the study showed that in vitro, when PlGF was applied early during human embryonic stem cell-CM differentiation, the production of CMs and ECs was enhanced. Finally, this comprehensive study confirmed the beneficial effects of PlGF in vivo on the development of human heart progenitor-derived cardiac muscle grafts in a mouse model. This study highlights the crucial role of PLGF in heart development, with potential implications for its application in cardiac angiogenesis therapeutic research [46] (Figure 1).

***Neuregulin-1 (NRG1):*** In heart failure, NRG1 activity increases as a compensatory mechanism to mitigate cardiac remodeling and disease progression [47]. In the heart, NRG1 plays a role in extracellular remodeling by inducing angiogenesis, promoting cardiomyocyte proliferation, and recruiting stem cells, all of which improve cardiac function [48]. NRG1 is primarily located in the microvascular and endocardial endothelium and plays a cardioprotective role against ischemic injury by inducing many downstream cascades, including extracellular signal-regulated kinase ½ (ERK1/2) mitogen-activated protein kinase, phosphoinositide 3-kinase (PI3-kinase), protein kinase B (AKT-kinase), mechanistic target of rapamycin (mTOR), and focal adhesion kinase (FAK) pathways. These pathways are key in promoting both hypertrophic and mitotic cardiomyocyte growth and enhancing cardiomyocyte resistance to apoptosis [49]. A recent study in mice investigated the role of NRG1-mediated activation of erythroblastic leukemia viral oncogene homolog ERBB4 receptors in cardiomyocytes in a transverse aortic constriction (TAC) model of heart failure. RNA sequencing of ECs showed that NRG1 influenced the expression of hypertrophic and fibrotic pathways, suggesting that NRG1/ERBB4 signaling modulates early hypertrophic and fibrotic pathways during cardiac remodeling [50] (Figure 1).

***Fibroblast Growth Factor (FGF):*** FGF plays an important role in the proliferation, migration, differentiation, and survival of ECs via autocrine/paracrine signaling pathways. Basic FGF (bFGF) and FGF2 bFGF attenuate myocardial injury by inhibiting apoptosis and promoting angiogenesis through a novel HIF-1α-mediated mechanism. The accumulation and activation of HIF-1α in the infarcted area upregulates bFGF expression, which enhances cardioprotective effects [51]. In a study that used an ischemia-reperfusion mouse model to study FGF2 deficiency, cardiac-specific overexpression of all four isoforms of human FGF2, and wild-type mice, demonstrated significantly higher recovery in post-ischemic function and reduced infarction size in mice with overexpressed FGF2, providing evidence that endogenous FGF2 plays a significant cardioprotective role against ischemia-reperfusion injury [52] (Figure 1).

***Transforming Growth Factor-β (TGF-β):*** During the conversion of fibroblasts to activated myofibroblasts in response to the inflammatory reaction caused by ischemic injury, transforming growth factor-β (TGF-β) is secreted into the ECM [7,53]. Secretion is crucial as cardiac fibroblasts possess the capability to undergo phenotypic alterations following an ischemic event, triggering an essential response and repair mechanism. This process leads to the formation of ECM components, which, in turn, contribute to the formation of myocardial scar tissue and facilitate the transition to myofibroblasts. These myofibroblasts play a vital role in maintaining the structural integrity of the ventricular wall [54]. Fibronectin serves as a stimulator of TGF-β, driving myofibroblast secretion [55]. Furthermore, in addition to this function, fibronectin may also contribute to cardiac angiogenesis. The heparin II domain of fibronectin binds to VEGF, thus potentially promoting angiogenic activity via VEGF-induced endothelial cell proliferation, migration, Erk activation, and neovessel formation in the infarcted area [56]. A study of bone marrow cells from humans with MI identified sulfatase SULF2 and its isoform SULF1 as endosulfatases that remove 6-O-sulfate groups from heparan sulfate (HS), eliminating docking sites for HS-binding proteins. These findings led us to hypothesize that Sulfs contribute to tissue repair following MI. To investigate this, a mouse model was used to study the effects of sulfatases. After coronary artery ligation in mice, the upregulation of Sulfs disrupted the interactions between VEGF and HS by removing HS-binding sites, thereby enhancing VEGF-mediated angiogenesis and promoting tissue repair [57] (Figure 1).

***Secreted Protein, Acidic and Rich in Cysteine (SPARC):*** SPARC is an ECM protein that significantly influences tissue remodeling and angiogenesis. Both VEGF and SPARC are involved in angiogenesis, with SPARC inhibiting the mitogenic effects of VEGF in human endothelial cells. This outcome was demonstrated in a study using a combination of in vitro experiments and biochemical assays to understand the effects of SPARC on VEGF-stimulated human microvascular endothelial cells (HMECs) [58] (Figure 1).

***Osteopontin (OPN):*** OPN is another matricellular protein that has a positive effect on cardiac angiogenesis. This was demonstrated in a small animal study that induced MI in wild-type and OPN knockout mice and concluded that the presence of OPN after ischemic injury is crucial for angiogenesis and may play a significant role in post-MI left ventricle remodeling in the context of myocardial angiogenesis [59] (Figure 1).

***Thrombospondins (TSPs):*** TSPs represent martricellar ECM proteins that serve a prominent role in the development of pathological processes of various cardiovascular diseases. TSP-1 and TSP-2 are the most multifunctional proteins due to their ability to interact with a variety of ligands with various abilities, including structural components of the ECM, cytokines, cellular receptors, growth factors, proteases, and other stromal cell proteins [60]. Specific to their angiogenic effects, TSP-1 and TSP-2 both interact with growth factors, cells, and the cardiac ECM. There have been multiple examples in vitro, in vivo, and in animal models that display TSP-1 as a downregulator of cardiac angiogenesis [61,62,63]. Studies investigating the same effect of TSP-2 suggest that it may exhibit an even greater ability to inhibit angiogenesis. Animal models and in vitro studies have shown that TSP-2 reduces microvascular endothelial cell proliferation and disrupts ECM-endothelial cell interactions [64,65] (Figure 1).

***Tenascins*:** Tenascins are another matricellular protein found in the cardiac ECM [66]. Of particular interest, tenascin-C (TNC) is one of the four types that is highly upregulated after changes in tissue organization structure, such as embryogenesis, inflammation, tissue repair, regeneration, or cancer invasion [67]. In the cardiac context, a study that utilized immunohistochemistry and RNA sequencing on lymphatic ECs gathered via human autopsy at various stages post-MI mapped the role of TNC in myocardial repair. The main findings of this study illustrated that TNC was upregulated during the inflammatory to early granulation phases post-MI in humans, thus influencing the transition from inflammation to tissue repair. Furthermore, the study showed that TNC may play a role in delaying the resolution of inflammation via macrophage polarization and inhibition of lymphangiogenesis, potentially leading to adverse remodeling in the chronic phase post-MI [66]. TNC has been shown to be angiogenic in myocardial repair after early stage MI in human samples [68]. Conversely, TNC has been shown to be embryonically angiogenic in the development of coronary arteries via PDGF-BB/PDGF-Rβ [69] (Figure 1).

***Periostin*:** Periostin is an osteoblast-specific factor (Osf-2) that functions as a regulator of cell adhesion and cell differentiation and is a crucial component in the organization of the ECM [70,71,72]. A mouse study exploring the role of periostin in heart disease demonstrated that in wild-type mice, a high-fat diet designed to induce metabolic syndrome significantly increased periostin levels along with aortic valve thickening, fibrosis, and MMP-2 and MMP-13 expression. These changes were markedly reduced in periostin knockout mice, indicating that periostin plays a critical role in the progression of heart disease [73]. Furthermore, outside of the realm of the heart, there has been a plethora of research implicating the role of periostin in tumor angiogenesis for a variety of cancers [74,75,76] (Figure 1).

**Table 1 biotech-14-00023-t001:** Comprehensive Overview of Regulators of Cardiac Angiogenesis.

ECM Component	Role in Cardiac Angiogenesis
Endostatin	Inhibits endothelial cell migration and proliferation [38] Upregulated in CAD, leading to diminished angiogenesis [39,40]
Vascular Endothelial Growth Factor (VEGF)	Promotes blood vessel growth and endothelial cell survival [43] Released by matrix metalloproteinases and other proteases during inflammation [43] VEGF-induced coronary angiogenesis promotes vascular health and contributes to improved tissue metabolism, which in turn supports cardiac function and perfusion in ischemic myocardium [41]
Placental Growth Factor (PIGF)	Regulates development of the vascular and lymphatic endothelium [42] Enhances VEGF tyrosine kinase activity, thus promoting new vessel formation [42] Induces nitric oxide production and influences cardiomyogenesis in progenitor cells [43]
Neuregulin-1 (NRG1)	Induces angiogenesis and cardiomyocyte proliferation [46] Protects cardiomyocytes from apoptosis [47] Activates pathways that promote cardiac hypertrophy and, therefore resistance to ischemic injury [47]
Fibroblast Growth Factor (FGF)	Impacts EC proliferation, migration, and survival [49] FGF2 promotes angiogenesis and cardioprotection [50] Enhances HIF-1α activity in response to ischemia, leading to increased recovery after ischemic injury [49]
Transforming Growth Factor-β (TGF-β)	Induces fibroblast to myofibroblast transition [52] Contributes to ECM remodeling and scar tissue formation [55] Promotes angiogenesis through interaction with fibronectin and VEGF [54]
Secreted Protein, Acidic and Rich in Cysteine (SPARC)	Inhibits VEGF-mediated endothelial cell mitogenesis [56] Involved in tissue remodeling during angiogenesis [56] Negatively regulates angiogenesis by limiting VEGF effects [56]
Osteopontin (OPN)	Promotes angiogenesis in post-MI ventricular remodeling [57] Important for left ventricle recovery after ischemic injury [57] Enhances neovascularization and myocardial repair processes [57]
Thrombospondins (TSPs)	TSP-1 and TSP-2 inhibit cardiac angiogenesis [62] Interact with ECM components and growth factors [59] Suppress endothelial cell proliferation and disrupt ECM to endothelial cell interactions [63]
Tenascins (TNC)	Upregulated post-MI during tissue repair [64] Inhibits lymphangiogenesis and modulates inflammation [64] May delay tissue healing and lead to adverse remodeling [64]
Periostin	Regulates cell adhesion and differentiation [68] Promotes EC tube formation and MMP secretion [71] Involved in angiogenesis both in cardiac degeneration and tumor progression [72]

## 6. Exploring Various Types of ECM Biomaterials

The cardiac ECM, which plays a crucial role in maintaining the structural integrity and function of the heart and is a regulator of cellular responses during cardiac injury, repair, and remodeling, presents certain features that allow scientists to conceive the potential for therapeutic interventions [77]. The strong confidence in the potential of cardiac ECM-derived therapies arises from observations in ischemic heart diseases, such as acute MI, where outcomes such as matrix degradation, disrupted deposition of new structural matrix, and disorganized scar architecture are associated with ECM dynamics [78].

Different biomaterials found within the ECM have been isolated and used to develop biocompatible scaffolds. These biomaterials are derived from a variety of sources, such as isolated human or animal tissues, isolated basement membrane matrices, cultured cell lines, and stem cells. These biomaterials are mostly composed of elastin, collagen, fibrin, and decellularized ECM [79,80].

*Collagen/Fibrin/Elastin-Based ECM Biomaterial:* As one of the ECM’s most essential roles is to provide the necessary structural support for the surrounding tissues, the necessary structural proteins must be present in the ECM scaffold, of which the most important are collagens [81,82]. Collagen-derived biomaterials have been utilized to study vascular cell behaviors, such as endothelial cell (EC) migration, proliferation, differentiation, and in vitro tubulogenesis [83,84,85,86,87]. Several studies have demonstrated the angiogenic potential of tandem collagen and fibrin hydrogels in the context of cardiovascular tissue and ECM [88]. Fibrin serves as the initial provisional matrix deposited universally immediately after injury, playing a crucial role in supporting the initial stages of tissue stabilization and regeneration [89]. Furthermore, fibrin’s inherent properties, such as counteracting the risk of immunological incompatibility and biodegradation, favor its use as an ECM protein in regenerative medicine [90]. Another major component of the ECM is elastin, which is composed of elastic fibers made up of tropoelastin subunits that are cross-linked with fibrillin microfibrils [91]. Its composition allows for compliance, resulting in what is referred to as elastin stretch, which plays an important role in blood vessels and lung tissue that undergo repeated physiological stretching [92]. Regarding the immunoreactivity of collagen- and fibrin-based ECM biomaterials, studies have shown that collagen-based materials influence immune cells, with hydrogel mechanics and adhesion domains playing significant roles in this immunomodulation. In contrast, fibrin materials induce both inflammatory and anti-inflammatory effects on the innate immune response and facilitate communication with the adaptive immune compartment, highlighting the potential of these biomaterials to enable precise control of immune responses in future therapies [93].

Multiple studies have attempted to configure the most effective ratio of ECM proteins in a scaffold to promote cardiomyocyte differentiation and function. One study used an integrated statistical model to create an optimized ECM formulation that augmented cardiomyocyte function in vitro [94]. Their 3D hydrogel was composed mostly of type 1 collagen (61%), laminin-111, and fibronectin in tandem with murine induced pluripotent stem cells. In vitro results showed that this statically produced array of proteins was optimal for the differentiation of cardiomyocytes from pluripotent stem cells [94]. Another study used neonatal rat cardiomyocytes cultured in an ECM scaffold made of type 1 collagen, gelatin, and fibronectin. The results showed that this solution was optimal for cardiac differentiation in Col-soft culture substrates, proving that this multi-culture substrate provides effective cardiac differentiation niches for regenerative medicine [95]. Another group of researchers blended the ECM matrix with fibrin and collagen in an attempt to engineer a functioning myocardium using human-induced pluripotent stem cell-derived cardiomyocytes. The study determined that the optimal density of fibrin and collagen in the matrix for optimal cardiomyocyte purity was a high concentration of fibrin and a low amount of collagen; however, functional beating tissue only occurred in pure collagen groups with a cardiac purity of >60%, illuminating the need for further studies to determine the optimal balance between each ECM protein [88]. A recent study focused on understanding the spatial distribution and microstructural morphologies of cardiac elastin in porcine left ventricles in an effort to better understand the role of cardiac ECMs in cardiac biomechanics. The results showed that when examining both the endocardial and epicardial layers, the combination of collagen and elastin reflected an optimal design, as collagen was more focused on mechanical strength, while elastin fibers facilitated recoil during systole. The main conclusion drawn is that future studies should fully understand the implications of elastin as a core biomaterial in the ECM as we attempt to develop bioengineered therapeutics [96].

*Decellularized ECM as a Biomaterial:* The application of decellularized extracellular matrix (d-ECM) is seemingly the most promising scaffold for inducing coronary angiogenesis as the mode of remodeling, replacing, and regenerating ischemic myocardium [97]. Decellurization of the ECM is a process that involves removing the full range of cells and cellular debris from a tissue or organ in an effort to isolate solely the ECM along with the components that comprise it. d-ECM is a means of bypassing the complexities that arise when attempting to synthetically produce the natural ECM and the scope of its components, and has shown some promise to be suitable for regenerative medicine applications. With the preservation of the complex architecture, d-ECM is further able to remove any immunogenic components that result in immune rejection [98]. Within the context of engineered cardiac and vascular tissues, d-ECM provides a promising potential for cardiac angiogenesis due to the preservation of the essential native bioactive molecules that foster homeostasis and facilitate tissue regeneration [99,100,101] (Figure 2).

While d-ECM offers advantages like preserving the structure of the respective tissues and organs, it also presents limitations. Factors such as the limited availability of tissue and organs, immune responses, risk of pathogens from allogeneic and xenogeneic tissues, arrhythmia from d-ECM injection, and the requirement of surgical implementation have prompted the development of alternative forms of d-ECM [102]. Furthermore, a study has shown that d-ECM preparation using sodium dodecyl sulfate (SDS)-based protocols for the decellularization process causes damage to the ECM microstructure and leads to increased thrombogenicity in vascularized transplantation and poor cell engraftment in recellularization procedures [103]. Another option involves cell culture-derived ECM, which can be utilized to create 3D scaffolds or combined with natural or artificial polymers for tissue engineering applications. Cell culture-derived ECM offers several advantages, including its ability to exist in pathogen-free environments, precise and controlled geometry, porosity, and optimal cell penetration [104,105] (Figure 2).

Another application of d-ECM is via tissue-derived ECM, which can be extracted from myocardial tissue and decellularized. Myocardial tissue-derived ECM has been shown to offer a complex array of biochemical and mechanical signals preserved from native myocardial tissue, which supports cell attachment, proliferation, and cardiovascular differentiation during subsequent recellularization [106]. Another study examined human cardiac patches using decellularized cardiac ECM and induced pluripotent stem cell-derived cardiac cells within the scaffold. An in vitro aspect of the study revealed normal contractile and electrical function, while in a rat AMI model, there was significantly improved cardiac function when the patch was applied to the infarcted area of the heart [107] (Figure 2).

## 7. Bioengineered ECM Therapies: Applications in Cardiac Angiogenesis

Bioengineered ECM therapies have garnered significant attention for their innovative approaches to restoring the function of the damaged myocardium. Techniques such as 3D hydrogels, bioprinting, and spatial patterning are being developed to replicate the native tissue architecture and create environments that promote cardiac angiogenesis. The basic functions, advantages, limitations, and applications of each ECM-based strategy are summarized in Table 2.

*3D Hydrogels:* ECM therapeutics commonly utilize hydrogel scaffolds, which are specialized groups of biocompatible 3D polymeric substances designed to mimic tissue properties. ECM hydrogels are injectable and compatible with various fabrication technologies, including 3D printing, microspinning, and electrospinning. Their versatility has significantly broadened the scope of clinical applications of ECM biomaterials [90]. By incorporating cells and promoting the degradation of non-viable tissue, these scaffolds support the growth of healthy tissue. Their capacity to retain moisture, maintain a porous structure, and adapt to changing sol-gel conditions has led to their widespread utilization. These qualities make hydrogel scaffolds effective tissue scaffolds, facilitating the exchange of cell metabolites and the removal of waste through their pores [108]. A crucial aspect in the design of ECM scaffolds is recognizing the specialized functions of the target vasculature that the biomaterials should emulate. Specifically, they should mimic tissue resilience and structure while effectively supporting cellular function [90]. The design of ECM-based hydrogels requires consideration of several factors. Mechanical properties are crucial, and various physical and chemical crosslinking methods must be utilized. Physical crosslinking methods, including ionic interactions, hydrogen bonding, and thermal induction-mediated solid-to-gel phase transitions, are essential [109]. Crosslinking conditions, such as light exposure sites, intensity, and duration, can significantly impact the mechanical properties of these hydrogels [110,111]. Moreover, attention must be paid to chemical properties, such as chemical crosslinking derived from chain growth polymerization. However, even with its versatility, unlike physical crosslinking, chemical crosslinking can involve toxic agents that react with bioactive compounds and produce deleterious effects [90] (Figure 3).

Some protein components of ECMs, like collagen and elastin, can naturally assemble into hydrogels, while other hydrogels can be composed of various synthetic polymers [112,113]. While being more difficult to assemble as compared to single-component hydrogels, the most successful 3D ECM hydrogels have been polymer-functionalized multi-component hydrogel networks of ECM molecules. They are preferred due to their ability to mimic the environmental complexities and biochemical and mechanical properties of the ECM in native tissues [114]. Furthermore, within the realm of 3D ECM hydrogels, d-ECM-based scaffolds have been an attractive option, specifically in the realm of angiogenesis, due to their structural ability to promote stem cell proliferation and differentiation [115,116] (Figure 3).

Numerous studies have demonstrated the potential of injectable cardiac ECM hydrogels as therapeutic options for IHD. A set of studies conducted using small (rat) and large (porcine) animal models showed promising outcomes with the injection of ventricular ECM hydrogels in an ischemia-reperfusion model. In the rat model, there was an increase in endogenous cardiomyocytes in the infarcted area, while cardiac function was maintained, and the risk of arrhythmias was reduced; meanwhile, in the porcine model, only the feasibility of trans-endocardial catheter injection was displayed [117,118]. In order to address both the structural and angiogenic necessities of a cardiac ECM-based therapeutic, a study used a rat model to employ a synthesized collagen-binding domain (CBD) peptide with a VEGF-mimic peptide (VMP). Post-AMI, angiogenesis was observed when CBD-VMP peptides were injected subcutaneously, but also when CBD-VMP peptides were injected intramuscularly in combination with an injectable cardiac ECM. Combined injection resulted in significant angiogenesis, reduced apoptosis, and recovery of cardiac function, providing evidence for the therapeutic potential of the combination of IM CBD-VMP/intracardiac-ECM injection as a clinical remedy [119]. A variation of injectable cardiac ECM-based therapeutics has shown similar results when derived with a foundation of ranging bioactives, such as VEGF, FGF, and derivations of it, hepatocyte growth factor (HGF) and derivations of it, among others [120] (Figure 3).

*Bioprinting:* In an effort to accurately reproduce the complex anatomical and cellular arrangements of cardiac tissue, biofabrication techniques, such as bioprinting, have proven successful [121,122]. 3D bioprinting techniques include inkjet, microextrusion-based, and laser-assisted bioprinting. Each regulates effective control of biomaterial disposition and comes with its own set of advantages and disadvantages [90]. The strengths of bioprinting are substantial, as it has been shown to support cellular viability, proliferation, maturation, and differentiation in constructed cardiovascular tissue [123]. Furthermore, its ability for structural mimicry also comes with bioprinted materials with the necessary chemical properties for proper function [124]. While 3D bioprinting is also capable of handling a large number of cells and is thus scalable, some limitations include the necessity of native human cardiovascular tissues/organs, which are limited in availability in order to precisely promote proper cell behavior and fabrication [125]. Similar to 3D hydrogels, d-ECM biomaterials are promising alternatives due to their ability to be engineered to replicate the capabilities of cardiovascular tissue. D-ECM bioink contains all the essential biochemical components of the native ECM that are crucial for cell proliferation and growth. Additionally, owing to their ability to be tailored to specific cell types and cellular mechanisms, d-ECM printed scaffolds provide the most potential for fabricating the necessary intricacies of bio-scaffolds within 3D structures [126]. Relevant to the cardiovascular sphere, 3D bioprinting in the decellularized context offers the ability to design cardiovascular tissues while maintaining their native structural and chemical components [127] (Figure 3).

A derivation of 3D bioprinting that has shown great potential as a carrier of cardiac ECM and associated angiogenic growth factors is bioprinted cardiac patches. Biocompatible polymers are the most widely used biomaterials for the construction of 3D tissues and organs; however, the biodegradable and rigid nature of these hydrogels does not sufficiently support the dynamics of cardiovascular tissue [128]. Furthermore, disadvantages include inflammatory reactions and pathological fibrotic states caused by polymer degradation [129]. To address these limitations, cell sheet engineering was developed to create a 3D structure by separating and layering 2D monolayers of cells that are grown without biodegradable materials and their associated issues, while also maintaining the angiogenic capacity of the scaffolds [130]. These engineered cardiac patches include living cardiomyocytes with angiogenic capacities to regenerate damaged or infarcted myocardium [131]. Research has focused on utilizing cardiac patches comprising native cardiac ECM, pediatric human cardiac progenitor cells (hCPCs), and gelatin methacrylate (GelMA) as epicardial devices that release paracrine factors in the injured myocardium. The summation of all of these factors onto the cardiac patch and its delivery to the heart resulted in improved differentiation and angiogenic potential within the rat myocardium compared to pure GelMA patches without hCPCs and cardiac ECM, indicating the groups reparative functionality [132] (Figure 3).

*Spatial Patterning:* Due to the unique composition and ultrastructure of each cell’s ECM, spatial patterning provides an alternative via the micropatterning of ECM components onto synthetic materials [133]. Photolithography or light-based patterning can be used to deposit differential ECM proteins onto a substrate [134]. While some of these techniques are unsustainable due to their cost and consequent scalability, alternatives such as elastomeric stamping techniques can provide a similar product while controlling the mechanism, incorporating microchannels and microfluids, and enabling the patterning of ECM components [135]. Furthermore, derivations such as electrospinning employ techniques that induce the proper alignment of nano- to microscale fibers [136,137]. Specifically, electrospinning techniques have proven valuable in ECM-based therapeutics targeting post-MI cardiac recovery due to the highly customizable substrate that is able to mimic the functionality of native ECM [138] (Figure 3).

An in vitro study was conducted to investigate the effect of a 3D poly-(ε-caprolactone) (PCL) nanofibrous scaffold molded via electrospinning methods on the differentiation of cardiomyocytes from iPSCs (induced pluripotent stem cells). This cocktail of components was designed as a way of mimicking the compositional and structural features of the native ECM in order to best produce engineered cardiac therapeutics. Among the many conclusions drawn from this study, the most significant regarding myocardial angiogenic potential is that this ECM mimicking scaffold holds promise for cardiac tissue regeneration and engineering [139] (Figure 3).

**Table 2 biotech-14-00023-t002:** Functions, Advantages, Limitations, and Applications of Bioengineered ECM-Based Strategies.

Component	Function	Advantages	Limitations	Applications/Outcomes
*3D Hydrogels*	− 3D biocompatible polymer scaffolds mimicking tissue properties [90] − Promote growth of healthy tissue by incorporating cells and degrading non-viable tissue [108] − Retain moisture and maintain a porous structure, making them effective as tissue scaffolds [108]	− Injectable and compatible with 3D printing, microspinning, electrospinning [117,118,119,120] − Facilitate cell metabolite exchange making effective tissue scaffold [108] − Mimic tissue resilience and structure [90]	− Mechanical properties depend on crosslinking methods [109] − Chemical crosslinking can introduce toxic agents [90]	− Used in cardiac ECM for myocardial infarction models [117,118,119,120] − Successful angiogenesis in small animal models [116,117] − Potential for cardiac regeneration and therapeutic applications [117,118,119,120]
*Bioprinting*	− Reproduces complex tissue structures using 3D printing techniques (inkjet, microextrusion, laser-assisted) [90] − Supports cellular proliferation and differentiation [123]	− Capable of handling large numbers of cells [125] − d-ECM bioink replicates cardiovascular tissue properties [126] − Scalable and structurally mimics native tissue [125]	− Limited availability of native cardiovascular tissues [128,129] − Biodegradable polymers can cause inflammation and fibrosis [128]	− Use of d-ECM bioink in angiogenesis [90] − Engineered cardiac patches with cardiomyocytes showing regenerative potential [131]
*Spatial Patterning*	− Micropatterning ECM components onto synthetic materials 18 March 2025 2:32:00 p.m. − Uses techniques like photolithography, electrospinning to control ECM structure [134]	− Customizable and capable of mimicking native ECM functionality [138] − Can be made of biocompatible materials − Effective for coronary angiogenesis [138]	− Photolithography is costly [134] − Challenges in scalability [135]	− Electrospun nanofibrous scaffolds show potential in cardiac regeneration [139] − Successful cardiomyocyte differentiation using iPSCs on ECM mimicking scaffold [139]

## 8. Future Directions and Concluding Remarks

Addressing the need for less invasive treatments for IHD remains a clinical priority. Researchers have devoted significant effort to uncovering the fundamental mechanisms responsible for the development and progression of IHD, identifying ECM as a major etiological factor. This has led to extensive efforts to elucidate the role of the ECM in both healthy and ischemic hearts, which have been shown to significantly enhance its therapeutic potential. To date, such work has been focused on delineating the various components of the ECM and their roles in the regulation of angiogenesis, paving the way for subsequent testing of biomaterial scaffolds capable of maintaining the cardiac structural integrity and function in the wake of injury.

Parallel investigations are ongoing to assess the most effective delivery methods for ECM-based scaffolds, incorporating such processes as 3D hydrogel construction, bioprinting, and spatial patterning. Recently, a study demonstrated the potential of 3D ECM particles derived from human MSCs. The model used LAD artery ligation to mimic IHD, followed by an intramyocardial injection of ECM, which resulted in reduced infarct size, enhanced capillary density, improved fractional shortening, and upregulated protein networks involved in cardiomyocyte contractility and fatty acid metabolism, all while showing minimal immune rejection and localized retention in the heart [140]. Due to such promising results and in an effort to optimize the translational potential of ECM modulations, our laboratory is conducting ongoing studies aimed at examining 3D ECM as a therapeutic option in a clinically relevant large animal model of IHD. The future directions of ECM-based therapies for cardiovascular diseases will require such translational preclinical studies to validate ECM efficacy, optimize delivery methods, and ensure safety, which may ultimately lead to human clinical trials.

## Figures and Tables

**Figure 1 biotech-14-00023-f001:**
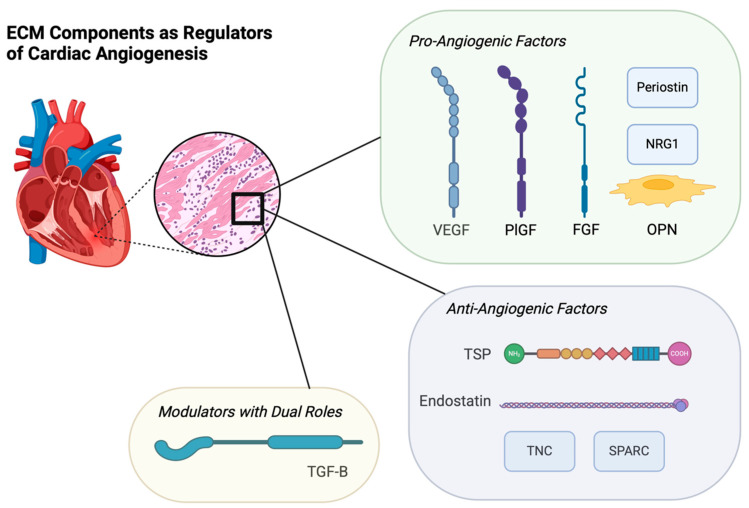
This figure highlights the roles of the different components of the ECM in regulating cardiac angiogenesis. The image depicts pro-angiogenic factors, such as VEGF, PIGF, FGF, Periostin, NRG1 and OPN, as well as anti-angiogenic factors like TSP, Endostatin, TNC, and SPARC. TGF-β has context-dependent effects on cardiac angiogenesis, acting as a promoter or inhibitor depending on the environment. Created in BioRender.com 9 February 2025.

**Figure 2 biotech-14-00023-f002:**
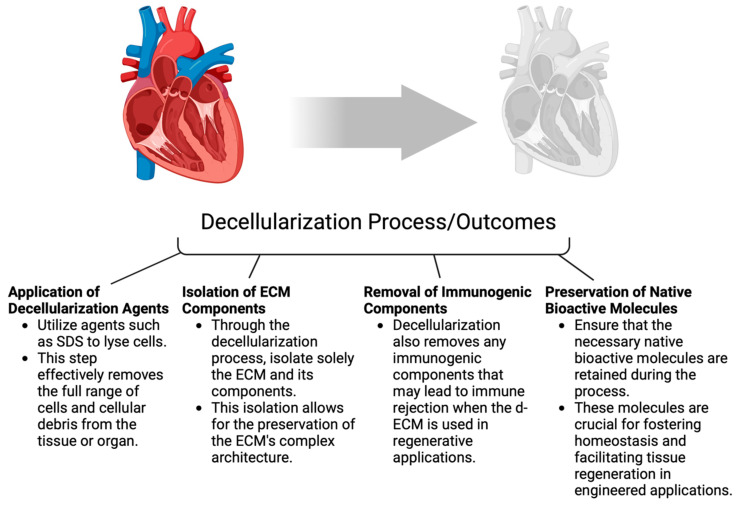
This diagram illustrates the decellularization process and its relevance to tissue engineering. The application of decellularization agents such as SDS effectively removes cellular components. The next step is the isolation of ECM components, where emphasis is placed on the importance of preserving the ECM structure. Following this, the removal of immunogenic elements is carried out to reduce the risk of immune rejection. Finally, great care is taken to preserve the native biomolecules that are crucial for maintaining homeostasis and supporting angiogenesis. Created in BioRender.com.

**Figure 3 biotech-14-00023-f003:**
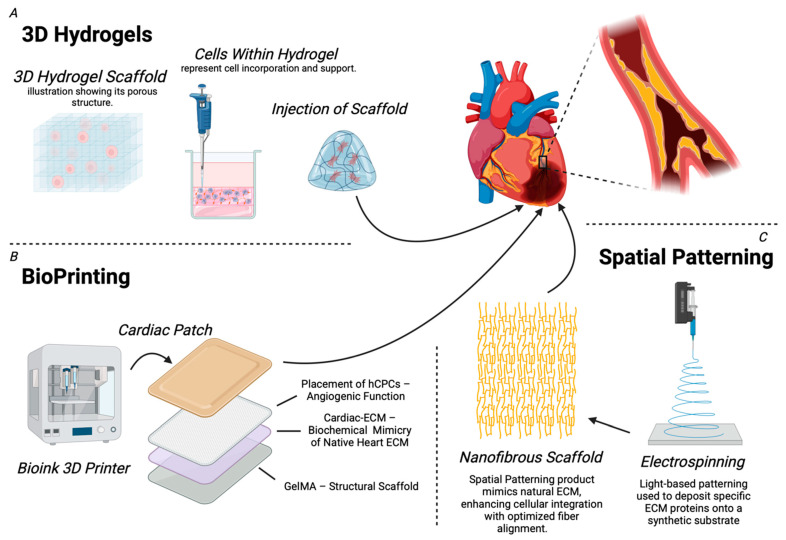
This diagram highlights advanced tissue engineering techniques aimed at regenerating damaged cardiac tissue to restore its functionality. (**A**) 3D hydrogels with their porous structure that support cellular incorporation and how the scaffold can be precisely injected into targeted areas are shown. (**B**) Bioprinting, particularly the fabrication of cardiac patches using bioink and 3D printing technology, along with the key components involved. (**C**) Finally, spatial patterning is shown, emphasizing how electrospinning is used to deposit ECM proteins onto a nanofibrous scaffold designed to replicate the natural ECM architecture. Created in BioRender.com.

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
