# Peer review of "Developments in Extracellular Matrix-Based Angiogenesis Therapy for Ischemic Heart Disease: A Review of Current Strategies, Methodologies and Future Directions"

_biotech, 2025, doi:10.3390/biotech14010023_

Round 1
Reviewer 1 Report
Comments and Suggestions for Authors
The authors present a review of the recent literature regarding the role of the ECM in cardiac function, how it is altered in disease and the unmet need for clinical interventions. The review is broad in scope but is logically structured and provides significant detail of each section. The work is well organised and a pleasure to read. I have no doubt that this will be well received with its target audience.
Comments on the Quality of English LanguageGood quality of English
Author Response
We would like to sincerely thank you for your thoughtful comments. We truly appreciate the time and effort you took to provide your valuable comments!
Comments 1: The authors present a review of the recent literature regarding the role of the ECM in cardiac function, how it is altered in disease and the unmet need for clinical interventions. The review is broad in scope but is logically structured and provides significant detail of each section. The work is well organised and a pleasure to read. I have no doubt that this will be well received with its target audience.
Response 1: Thank you for your thoughtful and encouraging feedback. We sincerely appreciate your kind words and are glad to hear that you found our review well-structured and engaging. Your support and insights mean a lot, and we hope the paper proves valuable to its readers. As no specific revisions were suggested, no changes were made based on the comments provided.

Reviewer 2 Report
Comments and Suggestions for Authors
- This review article focused on Extracellular Matrix-Based Angiogenesis Therapy for Ischemic Heart Disease, author emphasized with begin by providing an overview of the fundamentals of the cardiac ECM, focusing on the structural, functional, and regulatory mechanisms that drive its modulation. Subsequently, they explained the ECM’s interactions within both chronically ischemic and acutely infracted myocardium, emphasizing key ECM components and their roles in modulating angiogenesis. I am very appreciate that the manuscript clearly described the elements of the cardiac ECM are dynamic and have the capacity to continuously remodel, by which they can respond to physiologic changes. Because the ECM did play critical role in angiogenesis.
- Authors also point out the ECM in the Chronically Ischemic Heart was form with development of myocardial interstitial fibrosis (MIF)and the mechanism was different from ECM in the Acutely Infracted Heart
- Line 173-174.” These proteases cleave VEGF and can either promote angiogenesis in processes like carcinogenesis or suppress its angiogenic effects. “Could this sentence be explained more clearly?
- In section 5. Many angiogenesis factors critically involved in cardiac angiogenesis are mentioned, could VEGFRs be discussed?
- Since the composition of ECM in heart is different from other organs, usage of decellularized cardiac-ECM in induced pluripotent stem cell derived cardiac cells as the scaffold is necessary!
- The manuscript also mentioned the optimal density of fibrin and collagen in the matrix for the optimal cardiomyocyte purity was a high concentration of fibrin and low amount of collagen, indeed the increased level of collagen somehow reflects the level of cardiac fibrosis.
- Matricellular proteins are the non-structural proteins found in cardiac ECM which facilitate the spatial regulation of growth factor signaling and often impose a check on the fibrotic response comprised of the secretion of structural collagens by activated fibroblasts. As the infarct matures, cross linking of these collagens yields a detrimental increase in ventricular stiffness, with ongoing deposition culminating in progression to heart failure. I am very agree with these statement, thus the balance of proteolytic activity of the matrix metalloproteinases protein involved in extracellular matrix degradation must be precisely regulated by their endogenous protein inhibitors, the tissue inhibitors of metalloproteinases (TIMPs).
- Immune responses may lead inflammation; inflammation may cause local fibrosis, how about the immune reactivity of bioengineered ECM?
- Indeed, the most critical step is to develop effective delivery method for ECM-based scaffolds, incorporating such processes as 3D hydrogel construction, bioprinting, and spatial patterning.
- This article did provide new insight of the cardiac ECM as a major etiological factor for Ischemic heart disease, combine the knowledge of cardiac ECM and 3D scaffolds significantly enhance its therapeutic potential.
Author Response
We would like to sincerely thank you for your thoughtful and constructive comments. We truly appreciate the time and effort you took to provide valuable feedback, which has helped improve the clarity and depth of our manuscript.
Comment 1:
This review article focused on Extracellular Matrix-Based Angiogenesis Therapy for Ischemic Heart Disease, author emphasized with begin by providing an overview of the fundamentals of the cardiac ECM, focusing on the structural, functional, and regulatory mechanisms that drive its modulation. Subsequently, they explained the ECM’s interactions within both chronically ischemic and acutely infracted myocardium, emphasizing key ECM components and their roles in modulating angiogenesis. I am very appreciate that the manuscript clearly described the elements of the cardiac ECM are dynamic and have the capacity to continuously remodel, by which they can respond to physiologic changes. Because the ECM did play critical role in angiogenesis.
Response 1:
Thank you for your thoughtful feedback and kind words about our manuscript. We appreciate your recognition of the detailed discussion on the cardiac ECM and its role in angiogenesis. As no specific revisions were suggested, no changes have been made based on this comment.
Comment 2:
Authors also point out the ECM in the Chronically Ischemic Heart was form with development of myocardial interstitial fibrosis (MIF) and the mechanism was different from ECM in the Acutely Infracted Heart.
Response 2:
Thank you for your positive feedback. We appreciate your acknowledgment of our discussion on the differences in ECM remodeling between the chronically ischemic and acutely infarcted heart. As no specific revisions were suggested, no changes have been made based on this comment.
Comment 3:
Line 173-174: “These proteases cleave VEGF and can either promote angiogenesis in processes like carcinogenesis or suppress its angiogenic effects.” Could this sentence be explained more clearly?
Response 3:
Thank you for your suggestion. We agree that this sentence could be clarified for better readability. Therefore, we have revised it as follows:
"These proteases cleave VEGF, which can have dual effects depending on the context. In carcinogenesis, cleavage may release bioactive VEGF fragments that enhance angiogenesis, while in other conditions, degradation of VEGF can reduce its availability and suppress angiogenic signaling."
This revision can be found in red, on pg. 7 in the revised manuscript.
Comment 4:
In Section 5, many angiogenesis factors critically involved in cardiac angiogenesis are mentioned. Could VEGFRs be discussed?
Response 4:
Thank you for your thoughtful comment. We appreciate your suggestion and would like to clarify our reasoning for not discussing VEGFRs in detail. VEGFRs are inherently covered in our discussion of VEGFs on pages 6-7, as they are the receptors that mediate VEGF signaling. In Section 5, our primary focus is on the signaling factors themselves rather than the specific receptors. A detailed discussion of the full mechanistic scope of VEGF, including its receptors, would be an important addition but falls beyond the scope of this paper. Therefore, we have chosen not to make any changes based on this comment.
Comment 5:
Since the composition of ECM in the heart is different from other organs, usage of decellularized cardiac-ECM in induced pluripotent stem cell-derived cardiac cells as the scaffold is necessary!
Response 5:
Thank you for your comment. We agree that decellularized cardiac ECM plays a crucial role in supporting induced pluripotent stem cell-derived cardiac cells. This topic is already addressed in our discussion on pg. 14 where we highlight studies demonstrating the benefits of myocardial tissue-derived ECM as a scaffold, including its ability to support cell attachment, proliferation, and cardiovascular differentiation. Additionally, we discuss a study utilizing decellularized cardiac ECM with induced pluripotent stem cell-derived cardiac cells, showing improved cardiac function in a rat AMI model. Therefore, no changes have been made based on this comment.
Comment 6:
The manuscript also mentioned the optimal density of fibrin and collagen in the matrix for the optimal cardiomyocyte purity was a high concentration of fibrin and low amount of collagen. Indeed, the increased level of collagen somehow reflects the level of cardiac fibrosis.
Response 6:
Thank you for your comment. We agree that collagen levels are closely linked to cardiac fibrosis and that matrix composition influences cardiomyocyte purity (pg. 14). Our discussion highlights this with the role of fibrin and collagen concentrations in optimizing cardiomyocyte environments.
Comment 7:
Matricellular proteins are the non-structural proteins found in cardiac ECM which facilitate the spatial regulation of growth factor signaling and often impose a check on the fibrotic response comprised of the secretion of structural collagens by activated fibroblasts. As the infarct matures, crosslinking of these collagens yields a detrimental increase in ventricular stiffness, with ongoing deposition culminating in progression to heart failure. I am very agree with these statement, thus the balance of proteolytic activity of the matrix metalloproteinases protein involved in extracellular matrix degradation must be precisely regulated by their endogenous protein inhibitors, the tissue inhibitors of metalloproteinases (TIMPs).
Response 7:
Thank you for your thoughtful feedback. We are glad to hear that you agree with our discussion of the role of matricellular proteins and collagen crosslinking in heart failure progression. In response to your comment about the balance of matrix metalloproteinases (MMPs) and tissue inhibitors of metalloproteinases (TIMPs), we have added the following sentence to the manuscript to further clarify this regulation. This addition can be found in red, on pg. 6 in the revised manuscript.
Added text:
"This proteolytic activity is regulated by tissue inhibitors of metalloproteinases (TIMPs), which control the activity of MMPs and other ECM-degrading enzymes to prevent excessive matrix degradation or fibrosis."
Comment 8:
Immune responses may lead to inflammation; inflammation may cause local fibrosis, how about the immune reactivity of bioengineered ECM?
Response 8:
Thank you for your insightful comment. I had already discussed immune responses to decellularized ECM biomaterial in the manuscript. However, in response to your point regarding immune reactivity of bioengineered ECM, I have added a few sentences specifically discussing collagen/fibrin-based biomaterials and their associated immune responses. These additions can be found on pages 13-14, marked in red.
Added text:
"Regarding the immune reactivity of collagen- and fibrin-based ECM biomaterials, studies have shown that collagen-based materials influence immune cells, with hydrogel mechanics and adhesion domains playing a significant role in immunomodulation. In contrast, fibrin materials induce both inflammatory and anti-inflammatory effects on the innate immune response and facilitate communication with the adaptive immune compartment, highlighting the potential of these biomaterials to enable precise control of immune responses in future therapies"
Comment 9:
Indeed, the most critical step is to develop an effective delivery method for ECM-based scaffolds, incorporating such processes as 3D hydrogel construction, bioprinting, and spatial patterning.
Response 9:
Thank you for your insightful comment. I agree that developing effective delivery methods for ECM-based scaffolds is essential. No additional changes were made to the manuscript based on this comment, as the current version addresses the importance of scaffold development and delivery methods in relevant sections.
Comment 10:
This article did provide new insight of the cardiac ECM as a major etiological factor for Ischemic heart disease, combining the knowledge of cardiac ECM and 3D scaffolds significantly enhance its therapeutic potential.
Response 10:
Thank you for your kind words and positive feedback. I’m glad to hear that the manuscript has provided new insights into the role of cardiac ECM in ischemic heart disease and how combining it with 3D scaffolds can enhance therapeutic potential. No additional changes were made based on this comment.
